# Antibiotic Resistance of Selected Bacteria after Treatment of the Supragingival Biofilm with Subinhibitory Chlorhexidine Concentrations

**DOI:** 10.3390/antibiotics11101420

**Published:** 2022-10-17

**Authors:** Robin Früh, Annette Anderson, Fabian Cieplik, Elmar Hellwig, Annette Wittmer, Kirstin Vach, Ali Al-Ahmad

**Affiliations:** 1Department of Operative Dentistry & Periodontology, Faculty of Medicine, University of Freiburg, 79106 Freiburg, Germany; 2Department of Conservative Dentistry and Periodontology, University Hospital Regensburg, 93053 Regensburg, Germany; 3Department of Microbiology and Hygiene, Institute of Medical Microbiology and Hygiene, Medical Center, University of Freiburg, 79085 Freiburg, Germany; 4Institute of Medical Biometry and Statistics, Faculty of Medicine, 79104 Freiburg, Germany

**Keywords:** chlorhexidine, antibiotic resistance, antiseptic, biofilm

## Abstract

Due to increasing rates of antibiotic resistance and very few novel developments of antibiotics, it is crucial to understand the mechanisms of resistance development. The aim of the present study was to investigate the adaptation of oral bacteria to the frequently used oral antiseptic chlorhexidine digluconate (CHX) and potential cross-adaptation to antibiotics after repeated exposure of supragingival plaque samples to subinhibitory concentrations of CHX. Plaque samples from six healthy donors were passaged for 10 days in subinhibitory concentrations of CHX, while passaging of plaque samples without CHX served as control. The surviving bacteria were cultured on agar plates and identified with Matrix-assisted Laser Desorption/Ionization-Time of Flight-Mass spectrometry (MALDI-TOF). Subsequently, the minimum inhibitory concentrations (MIC) of these isolates toward CHX were determined using a broth-microdilution method, and phenotypic antibiotic resistance was evaluated using the epsilometertest. Furthermore, biofilm-forming capacities were determined. Repeated exposure of supragingival plaque samples to subinhibitory concentrations of CHX led to the selection of oral bacteria with 2-fold up to 4-fold increased MICs toward CHX. Furthermore, these isolates showed up to 12-fold increased MICs towards some antibiotics such as erythromycin and clindamycin. Conversely, biofilm-forming capacity was decreased. In summary, this study shows that oral bacteria are able to adapt to CHX, while also decreasing their susceptibility to antibiotics.

## 1. Introduction

The World Health Organization (WHO) declared that resistance to antimicrobials is an increasing global threat because the efficacy of antimicrobial therapy is severely compromised as a result of the lack of development of new classes of antibiotics [1]. The Review on Antimicrobial Resistance from 2016 predicted up to 10 million deaths connected to antimicrobial resistance by the year 2050—more than those caused by cancer [2]. Antibiotic resistance is not a novel phenomenon and some antibiotic resistance genes that were detected are thought to be 30,000 years old [3], thereby indicating that they existed long before the use and discovery of antibiotics as therapeutic agents. Compared to bacteria with resistance to individual antibiotics, multi-resistant pathogens which are no longer treatable pose a greater health risk [4]. Recently, there has been a discussion concerning a causal context between resistance toward antiseptics and cross-resistance toward antibiotics [5,6,7,8,9]. For instance, cross-resistances between antiseptics such as the quaternary ammonium compound (QAC) benzalkonium chloride and antibiotics have been described as being based on the mechanism of enhanced efflux [10], as well as the reduction in porins and the stabilization of the cell surface itself [11]. In the fields of dentistry and oral care, antiseptics are frequently used for oral biofilm control [12,13], with bisbiguanide chlorhexidine digluconate (CHX) being considered the gold standard [8,14]. Recently, it was shown by our group that microorganisms display the ability to adapt to antiseptics and are thereby able to also increase their biofilm formation. While major changes in antibiotic susceptibility could not be detected [15], the frequent treatment with subinhibitory CHX concentrations led to a genetic adaptation in the sense of the upregulation of efflux pumps in the bacteria [7].

However, these studies were conducted on planktonic bacteria in single cultures, while the oral cavity harbors more than 700 bacterial taxa which are organized as biofilms [16].

The growth of bacteria in biofilms differs significantly from planktonic growth and microorganisms can evolve genetically due to horizontal gene transfer [17]. In this context, biofilms are said to increase the resistance against environmental threats such as the increased concentration of antibiotics or antiseptics with the mechanism of horizontal gene transfer [18].

Furthermore, Mao et al. recently showed that multi-resistant phenotypes with additional resistance to CHX could be isolated from microcosm biofilms that were cultured in vitro and subjected to antiseptic treatment twice daily [19].

A threshold value of the MIC for defining the resistance against CHX has not yet been established [20], but the adaption of oral bacteria to the antiseptic has been observed [15]. Different mechanisms of action are discussed for such adaptations, including changes in the membrane that complicate the penetration of CHX into the inside of the cell and the expression of multidrug efflux pumps [21]. Such adaptions could be promoted by the mutagenicity of substances. Although it has not yet been clarified whether CHX possesses mutagenic effects, CHX was highlighted as being the only bisguanide that showed increased mutagenicity [22]. Furthermore, different studies have shown an increased mitotic crossing that is over-induced by the presence of CHX [23] and, in animal experiments, there were changes in cells from the blood and the kidneys after treatment with CHX [24]. Nevertheless, there are experiments showing that there is negative genotoxic potential CHX [25].

At present, there is too little knowledge to assess the risks associated with the widespread use of CHX in dentistry with regard to selection for antibiotic-resistant phenotypes or the development of cross-resistances to antibiotics.

Therefore, the present study aimed to investigate the phenotypic adaptation of bacteria from supragingival plaque to CHX upon multiple exposures to subinhibitory concentrations in vitro. Furthermore, cross-resistance to antibiotics and biofilm-forming capacity were assessed.

## 2. Materials and Methods

### 2.1. Study Design

Supragingival plaque samples were collected from six healthy volunteers (P1–P6) who had been recruited in an earlier clinical study approved by the ethical committee of the University of Freiburg (604/16). To be included, the test persons had to be aged between 18 and 80, be non-smokers, do not suffer from systemic diseases or conditions, display periodontal and dental health (no decayed, missing, or filled teeth), and have not used antibiotics in the last six months. After 24 h of abstaining from oral hygiene measures, plaque samples were taken from different teeth, pooled in 1.5 mL reduced transport fluid (RTF), and stored at −80 °C.

### 2.2. Passaging of Supragingival Plaque with Subinhibitory Concentrations of CHX and Isolation of Bacteria

Supragingival plaque samples were thawed in a water bath at 36 °C. Afterward, a dilution series (10^−1^ to 10^−3^) was prepared in Peptone-Yeast-Bouillon (PY) (Merck, Darmstadt), and aliquots from each dilution step were plated on Columbia blood agar (CBA) plates and incubated for five days at 36 °C under aerobic conditions (5% CO_2_). Colonies were quantified visually, discriminated according to their respective colony morphology and hemolysis behavior, and separated on fresh CBA plates. These pure cultures were subsequently identified using matrix-assisted laser desorption/ionization time-of-flight mass spectrometry (MALDI-TOF MS) on a MALDI Biotyper^®^ sirius (Bruker, Billerica, MA, USA), as described previously [26]. The bacteria were stored at −80 °C.

A 96-well-microtiter-plate (Greiner bio-one, Frickenhausen, Germany) was inoculated with 200 µL brucella broth (BBF) (Becton Dickinson, Francis Lakes) containing CHX (GlaxoSmithKline, London-Brentford, GB) in concentrations from 0.02% to 1.5 × 10^−5^% CHX to reach a volume of 100 µL per well. Subsequently, 10 µL of the 1:100 diluted plaque sample, which was defrosted before, was added to each well, and thoroughly mixed by pipetting up and down. The well plates were incubated for 24 h at 36 °C under aerobic conditions (5% CO_2_). Afterward, the minimum inhibitory concentrations (MICs) were assessed and the bacterial population from the well with the highest antiseptic concentration still exhibiting bacterial growth (Sub-MIC) was used for the inoculation of the next passage. The content of the Sub-MIC well was used as the inoculum for a newly prepared microtiter plate and subsequently, the next passage of MIC evaluation and growth was conducted as described previously. This procedure was executed for 10 passages (P1 to P10). At the end of passaging, each well was plated on CBA and incubated for five days at 36 °C under aerobic conditions (5% CO_2_). These experiments were conducted in quadruplicate. Cultures were identified by MALDI-TOF MS as described above and stored at −80 °C.

### 2.3. Minimum Inhibitory Concentrations (MICs) of Bacterial Isolates before and after Passaging with CHX

For evaluation of the effects of passaging supragingival plaque samples in subinhibitory concentrations of CHX, MICs of the isolates before and after passaging were determined for CHX. These isolates were subsequently cultivated on Columbia blood agar (CBA) plates for 24 h at 36 °C in an aerobic atmosphere with 5% CO_2_.

The assorted bacteria were cultured from their stocks at −80 °C in Barnes glucose phosphate medium mixed with 15% glycerol. Afterward, the colonies were suspended in NaCl (0.9%) yielding 0.5 MacFarland (corresponding to approximately 1 × 10^8^ CFU/mL). BBF served as a culture medium for the MIC-Testing. The suspension was diluted 1:10 in BBF (1 × 10^7^ CFU/mL) and 5 µL of the dilution were inoculated in a 96-well-microtiter-plate (Greiner bio-one, Frickenhausen, Germany) with 100 µL of BBF so that the final concentration of the inoculum was 5 × 10^5^ CFU/mL. The concentration of 5 × 10^5^ CFU/mL was ensured with an inoculum control.

The MIC-Testing was conducted with eight concentrations from 0.02% CHX (1:10) to 1.5 × 10^−5^% CHX (1:1280). The MIC was defined as the lowest concentration of CHX at which bacterial growth inhibition was apparent [27]. A total of 10% of the volume of the wells was spread out on CoBl and incubated for five days at 36 °C and 5% CO_2_ after which the cultures were counted, and the minimum bactericidal concentration (MBC) was identified as the well in which 99.9% growth was inhibited. All tests were performed sixteenfold.

### 2.4. Biofilm Forming Capacity

The biofilm-forming capacity of the asserted bacteria isolated before and after passaging in CHX was determined, as described in detail earlier [28,29]. The asserted bacteria were cultured from their stocks at −80 °C in Barnes glucose phosphate medium mixed with 15% glycerol. Afterward, the colonies were suspended in NaCl (0.9%) yielding 0.5 MacFarland (corresponding to about 1 × 10^8^ CFU/mL). Tryptic-soy-broth (TSB) served as a culture medium for the biofilm assay, which was conducted using a microtiter plate assay [29,30]. The suspension was diluted 1:10 in TSB (1 × 10^7^ CFU/mL). A total of 5 µL of the dilution were inoculated in a 96-well-microtiter-plate (Greiner bio-one, Frickenhausen, Germany) with 100 µL of TSB so that the final concentration of the inoculum was 5 × 10^5^ CFU/mL. The concentration of 5 × 10^5^ CFU/mL was ensured with an inoculum control.

Subsequently, the wells of polystyrene 96-well tissue-culture plates (Greiner bio-one, Frickenhausen, Germany) were filled with 100 µL of the diluted isolates and the diluted antiseptic (final concentration of 0.02%–1,5 × 10^−5^% CHX). The plates were incubated for 24 h at 36 °C in an aerobic atmosphere with 5% CO_2_ after which 100 µL of medium were changed and the plates were incubated for another 24 h. After draining the culture medium, the 96-well-plates were washed three times using 300 µL NaCl to remove the non-adherent bacteria and air-dried for 30 min. The adherent microorganisms that remained were stained with 0.1% gentian-violet (Carl Roth GmbH, Karlsruhe, Germany) for 10 min. Excessive dye was removed by rinsing the plates with distilled water. Afterward, the plates were dried for 60 min at 36 °C. The dye was resolubilized by the addition of 50 µL of absolute ethanol (99.9%) (Honeywell, Muskegon, USA) for a density check in each well. The optical density was measured with a Tecan Infinite-M200Plate-Reader (Tecan, Crailsheim, Germany) and proceeded at a wavelength of 595 nm (OD595). All tests were conducted sixteenfold and the mean values were detected.

### 2.5. Antibiotic Susceptibility Testing

Antibiotic susceptibility was tested using the epsilometertest (Etest). The following thirteen antibiotics were tested: erythromycin, clindamycin, penicillin G, ampicillin, gentamicin, tetracycline, vancomycin, meropenem, ciprofloxacin, cefuroxime, tigecycline, moxifloxacin, and amoxicillin. For the testing, an overnight culture of the isolate was suspended in 0.9% NaCl to a turbidity of 0.5 McFarland. Thereafter, Müller-Hinton with horse blood Plates (MHPBM) were inoculated for *Streptococcus oralis* isolates and brucella blood agar plates (BBA) for *Granulicatella adiacens* isolates. The testing strips were put on the plates, which were then incubated for 24 h at 36 °C and 5% CO_2_. The MIC of the antibiotics was determined where the ellipse was cutting the scale and tests were repeated when the MICs were different by two or more levels.

### 2.6. Statistical Analysis

For descriptive analyses, the mean, median, and standard deviation were computed. Bar charts were used for the graphical presentation of the results.

The range of values was previously tested for distribution. With the distribution classified as nonparametric, the statistics were performed as a non-parametric calculation.

The Kruskal–Wallis-Test was used for each state of passage to check for differences between concentration groups concerning the biofilm values as well as the MIC and MBC. For pairwise tests in subgroups, the Wilcoxon rank-sum test was applied. For the antibiotic resistance analysis of *G. adiacens*, the t-test was used because a non-parametric test does not have enough power due to the low case numbers. For multiple statistical tests, the Bonferroni correction was used.

The calculations were performed with the statistical software STATA 17.0 (StataCorp, College Station, TX, USA) and the charts were generated using GraphPad Prism 9.2.0 (GraphPad Software, San Diego, CA, USA).

## 3. Results

### 3.1. Selection of Isolates by Passaging in Sub-Inhibitory Concentrations of CHX

After passaging the samples of supragingival plaque for 10 days in subinhibitory CHX-concentrations, only bacteria identified as *Streptococcus oralis* were found in the biofilms of all six samples. Another isolate that was not consistently found except in one of the experiments was identified as *G. adiacens*. The isolates of *G. adiacens* after treatment showed an increased MIC of CHX by four levels (from 0.00062% CHX to 0.0025% CHX) and MBC by two levels (from 0.0025% CHX to 0.005% CHX) in comparison to the original isolates (Figure 1). In contrast to the original before treatment with subinhibitory CHX concentrations and to the growth control passaged without CHX, the *S. oralis* isolates gained a reduced susceptibility to CHX by two (from 0.0025% CHX to 0.005% CHX) and (from 0.0025% CHX to 0.01% CHX) four levels with regard to their MIC and MBC, respectively. The original isolates and the isolates from the growth control both showed the same susceptibility so that an effect of the subinhibitory CHX concentrations could be presumed (Figure 2). The exact rating-values of the MALDI-TOF-analysis are given in the Appendix A.

### 3.2. Biofilm-Forming Capacity

The biofilm-forming capacity was investigated without CHX and under the influence of various subinhibitory CHX concentrations. The original isolates before passage, as well as the original isolates after passaging without CHX, and the CHX-passaged isolates were examined. For *S. oralis* isolates, no significant differences in biofilm formation occurred by the 10-day passage without CHX compared to the original isolate. For the concentration of 0.00032% CHX, 0.00015% CHX, and 0% CHX no statistically relevant differences were detected. The isolate passaged 10 times (for 10 days, one day for each passaging) in CHX showed no significant change in response to a CHX concentration of 0.00032% in comparison to the original isolate. However, it showed a significant decrease in biofilm formation as compared to the original germ for 0.00015% CHX (*p* = 0.0002) and 0% CHX (*p* = 0.00001). When comparing the CHX-treated isolate to that of the growth control, there were no significant differences in biofilm formation for the 0.00032% CHX and 0.00015% CHX (*p* = 0.2420) concentrations, but for biofilm formation without CHX addition, the values of the growth control isolate were higher. In summary, isolates after CHX-treatment showed a tendency of being limited in their biofilm formation capacity (Figure 3).

*G. adiacens* showed a slight increase in the biofilm formation capacity after CHX-passaging compared to the untreated bacterium. Due to the small sample numbers, it was not possible to define how *G. adiacens* in general is changing its biofilm building capacity after CHX-treatment (Figure 4).

### 3.3. Antibiotic Susceptibility Testing

When evaluating the Etests, differences depending on the passaging (with and without CHX) of the isolates were discovered. The CHX concentrations from which the isolate was obtained did not influence the phenotypic antibiotic resistance patterns. The susceptibility of CHX-passaged *S. oralis* isolates towards erythromycin decreased significantly in comparison to that of the original isolates (*p* = 0.0016). The MIC of clindamycin also showed a significant increase for the CHX-treated isolates compared to the untreated ones (*p* = 0.001). An increase in the MIC values was also shown for amoxicillin (*p* = 0.0118) and ampicillin (*p* = 0.0156). For the comparison of the initial isolate and the isolate after passaging without CHX, there were significant diminutions of the susceptibility for the antibiotics amoxicillin (*p* = 0.0159) and ampicillin (*p* = 0.0260). Between the individual sample throughputs, major differences were detected in the direction of increased as well as in the direction of decreased sensitivity for tetracycline (Figure 5 and Figure 6). *G. adiacens* isolates exhibited lower susceptibility to erythromycin (*p* = 0.00001) and clindamycin (*p* = 0.00001). Higher MICs also appeared for penicillin G (*p* = 0.0497), tetracycline (*p* = 0.0075), cefuroxime (*p* = 0.0014), and ciprofloxacin (*p* = 0.0227) (Figure 7). The actual values of the Etests have been added in the Appendix A.

## 4. Discussion

Because of the widespread use of antiseptics such as CHX in dental practice and oral care products, there is increased clinical interest in whether bacteria in biofilms can adapt to CHX and if this adaptation is accompanied by the development of cross-resistance to antibiotics. Hence, the present study simulated the frequent exposure of dental plaque bacteria to CHX by passaging biofilm samples in subinhibitory concentrations of CHX. After the cultivation of the surviving microorganisms, the bacterial colonies visible on the plates were classified according to chemical and morphological criteria and transferred into pure cultures. Such modified microdilution methods have been used by other authors to investigate the adaption of bacteria to antiseptics after frequent exposure to subinhibitory concentrations [6,9,15,31,32].

The limitations of determining the bacterial diversity of natural niches on agar plates have been reported for the culture technique in general [33,34]. In the present study, the intensive treatment of the plaque samples with subinhibitory concentrations of CHX greatly reduced the bacterial diversity. However, if the phenotypic expression, virulence factors of various bacteria, and their resistance towards antimicrobials are to be studied, the culture technique is the method of choice [35]. MALDI-TOF-MS was used for the identification of bacterial isolates depicted in the present study. This method is characterized by short analysis times and high precision at low cost while being easy to use [26]. Mass spectrometry identifies bacteria deposited in the database with high confidence, regardless of the culture medium used. Hence, MALDI-TOF-MS can be used to replace the time-consuming biochemical identification approach [36]. To verify phenotypic adaptions of the bacteria regarding their susceptibility to CHX, the species were isolated and spread out again in an antiseptic-free nutrition medium. Subsequently, the bacteria were tested to determine their MIC for CHX. The adaptions found in the microdilution assay could be reproducibly verified in the MIC testing. Although no exact threshold values exist to define the resistance towards antiseptics to date [37], Chapman defined MIC increases by a factor of at least four as clinically relevant [38]. In the present study, bacteria that survived the treatment with CHX and gained at least a fourfold increase in their MICs to CHX (*S. oralis* and *G. adiacens*) were both Gram-positive cocci. Other Gram-positive cocci such as *Enterococcus faecalis* were also shown to develop resistance against CHX in previous studies [39]. Furthermore, we recently showed that twice-daily treatment of microcosm biofilms with CHX led to an ecological shift toward Gram-positive cocci in these biofilms. [19]. *S. oralis* was reported to have lower environmental requirements for a high growth rate in combination with a high adherence ability [40,41]. This allows *S. oralis*, as an initial colonizer, to take advantage of the selection pressure of the environment and overgrow other germs. Additionally, in the literature, *S. oralis* was considered to have the ability to grow under coaggregation, which favors the development of resistance [42,43]. Coaggregation enables *S. oralis* to gain firm adhesion in a very short time, which gives it a growth advantage over other oral bacteria [44]. The effect of coaggregation is also discussed for *G. adiacens* and possibly provides a higher initial tolerance to antiseptics and the associated development of resistance [45]. Considering that both *S. oralis* and *G. adiacens* are under suspicion of being co-triggers for endocarditis [46,47] and that *S. oralis* appears to induce sepsis under certain conditions [48], the resistance of corresponding isolates poses a high general medical risk. However, acquired resistance from in vitro experiments could not necessarily be connected to resistance in vivo. At the end of the CHX selection, both bacterial species grew at low CHX concentrations in the range of 0.005% to a maximum of 0.01%, which are much lower than the therapeutic dosage of 0.2% in which CHX has usually been used. In this respect, it can be assumed that it is improbable for these bacteria to grow at the therapeutically applied concentrations of CHX. On the other hand, it could not be excluded that areas with subinhibitory CHX concentrations exist within the oral biofilm due to low diffusion caused by the extracellular substances when bacteria grow in biofilms [8]. Such areas may lead to the adaptation of some oral bacteria and subsequently to the development of resistance to CHX. Hence, the growth of bacteria in biofilms and resistance against antiseptics can synergistically influence each other [49]. It should be emphasized that bacteria within biofilms are highly protected against the host immune system and antimicrobials [40,48].

CHX is considered the gold standard to control the growth of oral biofilm. However, the effects of sub-inhibitory concentrations of CHX on the adaptation capabilities of oral bacteria have not been investigated sufficiently to date. A recent study showed that sub-inhibitory concentrations of antibiotics as a stress factor can increase the biofilm formation capacity of clinical *Enterococcus faecalis* isolates [28]. Frequent subinhibitory CHX-treatment was also shown to increase the synthesis of glucan in *S. mutans*, thereby promoting biofilm formation [7].

Such sub-inhibitory concentrations have been assumed to exist in deeper layers of the oral biofilm as was discussed by Cieplik et al. This could be the cause for the results of the present study which showed survival of *S. oralis* and *G. adiacens* after treating the dental plaque with different CHX concentrations in vitro. Moreover, in the present study, such sub-inhibitory CHX concentrations may have led to the fact that isolates treated with CHX showed reduced biofilm formation compared to pretreated isolates and isolates of the growth control. The frequent presence of subinhibitory CHX-concentration is not only proposed to influence biofilm building capacity but also to induce cross-resistance against antibiotics [8,20].

From the results of the present study, it can be inferred that repeated CHX treatment as well as the passaging itself (without CHX) can induce major changes in the resistance patterns of antibiotics. These adaptabilities can evolve in either direction of a higher or a reduced susceptibility. A closer look at antibiotic susceptibility test results reveals that altered resistance patterns to the macrolide erythromycin and the lincosamide clindamycin occurred for *S. oralis* after pretreatment with CHX. The change in resistance to erythromycin and clindamycin can be explained by the fact that lincosamides and macrolides have a comparable molecular mechanism of action. Thus, if the pattern of resistance to either antibiotic changes, it can be assumed that there is a change in the pathway that also influences the mode of action for other antibiotics with similar mechanisms of action. This can either be a mutation of the genes coding for the rRNA methylases, resulting in general resistance to macrolides and lincosamides, or an enzyme that modifies the binding site for the two antibiotics [50]. In the literature, adaptations caused by efflux pumps are discussed for CHX-induced antibiotic resistance; this is also a possible mechanism for the development of cross-resistance to clindamycin and erythromycin [5]. In order to be able to make a differentiated statement about the mechanism of resistance development, an analysis of the genome of the bacterial strains would be conceivable in future studies.

For *G. adiacens*, a large shift in antibiotic resistance values was shown, whereby the resistance of *G. adiacens* to some antibiotics such as penicillin, erythromycin, or clindamycin has also been described in the literature [47]. However, resistance that spans a very broad band of antibiotics as depicted in the present study has not been described and could be a consequence of the intensive treatment with CHX which led to the selection of a resistant *G. adiacens* isolate. As a mechanism for such changes in antibiotic resistance, the expression of efflux pumps and changes in the binding structures of the cytoplasmic membrane should be mentioned [51]. These adaptation mechanisms could also be due to genetic adaptation resulting from the potential mutagenicity of CHX [24] which should be investigated in future studies in this regard. If such a mutagenic effect of CHX does exist, then antibiotic sensitivity could increase or decrease after CHX treatment. Based on a possible mutagenic effect and the resulting increased gene variation, as already shown by other authors [23], it can be assumed that the development of cross-resistance is promoted. However, resistance patterns for isolates of the untreated growth controls were also altered. These changes could be associated with the passaging itself and may be explained by the fact that the bacteria have a relatively short generation time causing natural genetic changes which could also lead to alterations in the level of antibiotic sensitivity [52]. The resistance breeding accordingly spans about 240 generations with a duration of 10 days. This high number of generations also allows random genetic adaptations to occur. To verify the CHX-induced adaptions in biofilm building capacity and cross-resistance to antibiotics, studies should be conducted with a larger number of different oral bacterial species, whereby including oral bacterial strains from patients who underwent intensive CHX therapy could add some new insights to this topic. Nevertheless, although CHX appears to retain its biofilm-reducing effect, the use of alternative compounds such as photodynamic therapy or natural compounds should be considered to avoid antimicrobial resistance in the future [53,54].

## 5. Conclusions

The results from the present study suggest that frequent treatment with CHX can lead to adaptations of some oral taxa to CHX, in addition to changes in antibiotic susceptibilities, while biofilm formation capacity is not affected. Due to the fact that CHX is frequently used as the “gold standard” in dentistry, the exact influence of CHX on the antibiotic sensitivity of oral microorganisms should be evaluated with a larger number of oral bacterial species. In vivo experiments are also required to evaluate the influence of CHX on the development of resistance toward antiseptics and antibiotics after clinical use.

## Figures and Tables

**Figure 1 antibiotics-11-01420-f001:**
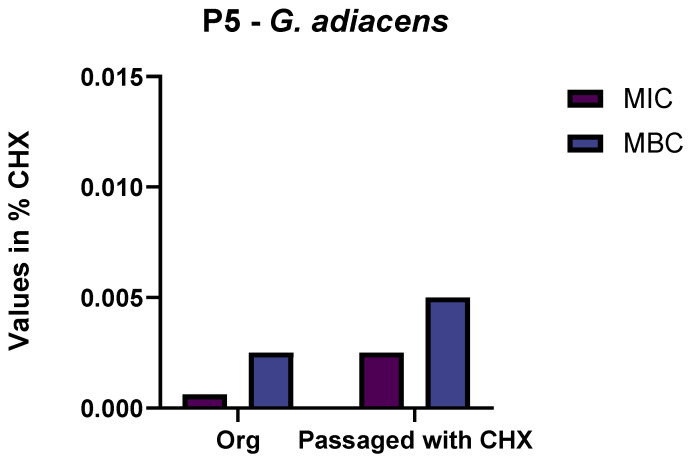
Minimum inhibitory concentration (MIC) and minimum bactericidal concentration (MBC) of *G. adiacens* isolated before (Org) and after passaging, with and without the addition of chlorhexidine (CHX). Values are depicted in percent of CHX. P5: Proband 5.

**Figure 2 antibiotics-11-01420-f002:**
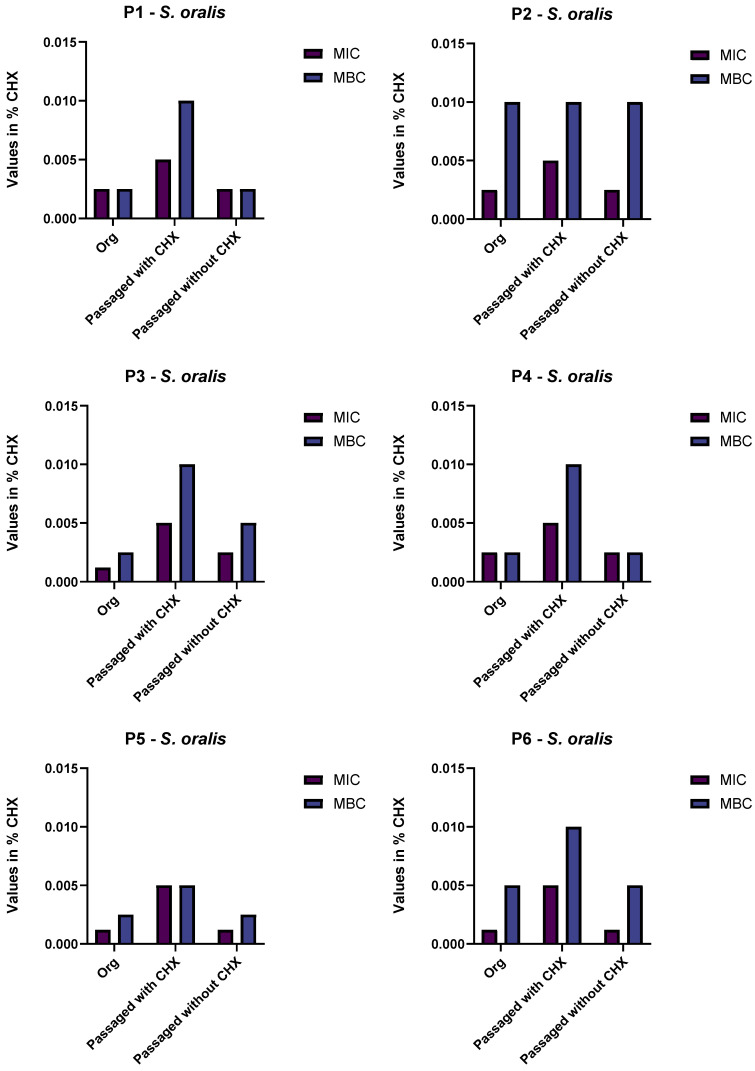
Minimum inhibitory concentration (MIC) and minimum bactericidal concentration (MBC) of the original *S. oralis* isolate and the isolates after passaging with and without the addition of chlorhexidine (CHX). Values are depicted in percent of CHX. P1–P6: Probands 1–6.

**Figure 3 antibiotics-11-01420-f003:**
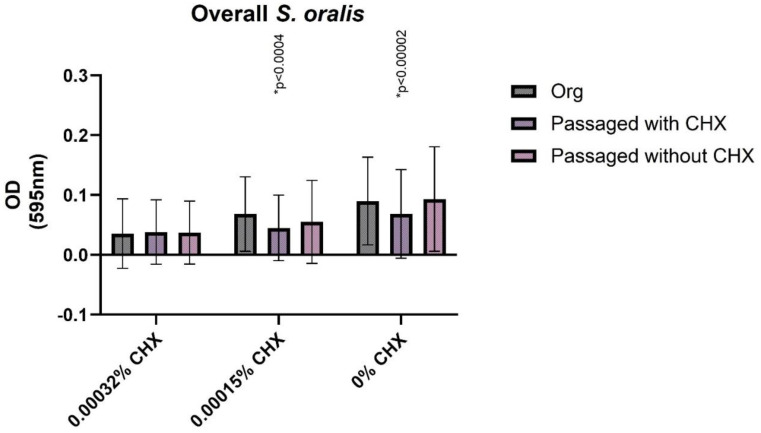
Biofilm formation at different subinhibitory concentrations of chlorhexidine (CHX). Graphical illustration of the biofilm formation (OD values) of *S. oralis* isolates exposed to subinhibitory concentrations of CHX. The percentages of CHX are shown on the *X*-axis, and the OD values are shown on the *Y*-axis. *p*-values in comparison to the original isolate (* significant difference).

**Figure 4 antibiotics-11-01420-f004:**
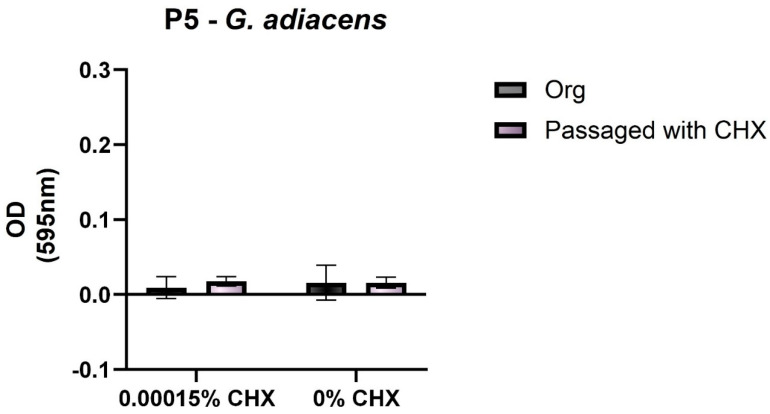
Biofilm formation at different subinhibitory concentrations of chlorhexidine (CHX). Graphical illustration of the biofilm formation (OD values) of *G. adiacens* exposed to subinhibitory concentrations of CHX. Percentages of CHX are shown on the *X*-axis, and the OD values are shown on the *Y*-axis. P5: Proband 5.

**Figure 5 antibiotics-11-01420-f005:**
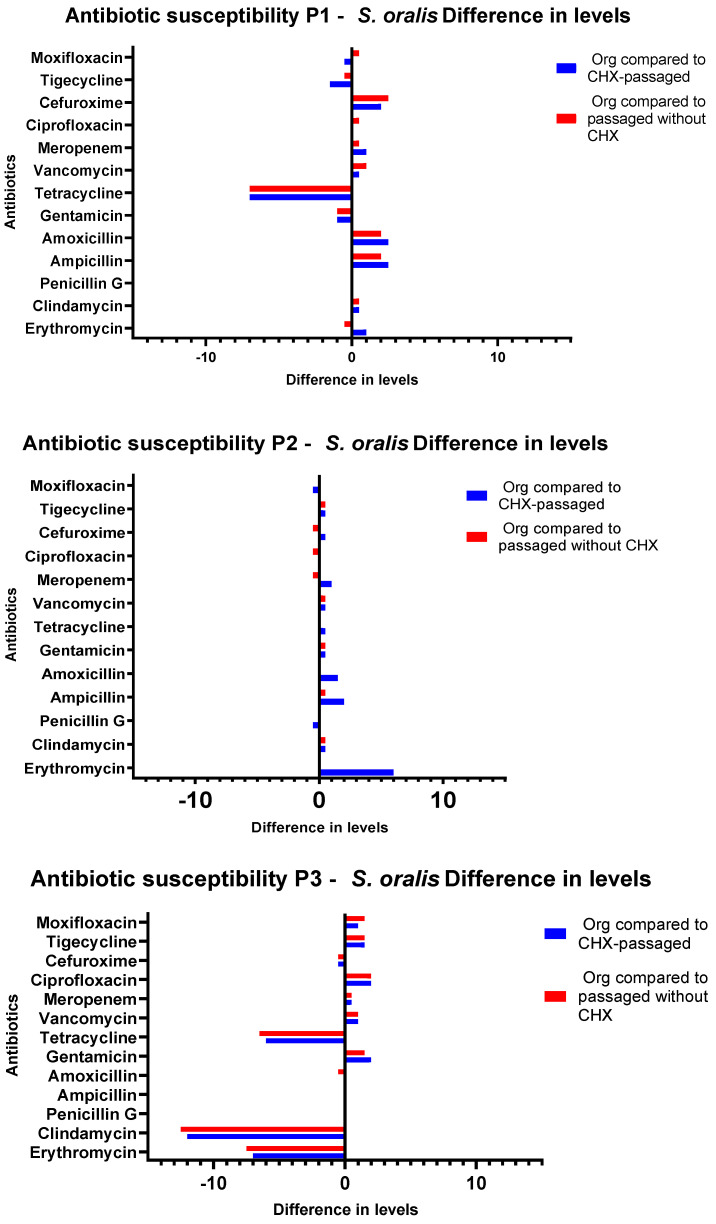
Antibiotic susceptibility testing of *S. oralis* isolates from Subject 1–Subject 3. Graphical depiction of the difference in susceptibilities in levels between the original isolate and the isolates after passaging with and without CHX. The levels are drawn on the X-axes, and the antibiotics are shown on the Y-axes. P1–P3: Probands 1–3.

**Figure 6 antibiotics-11-01420-f006:**
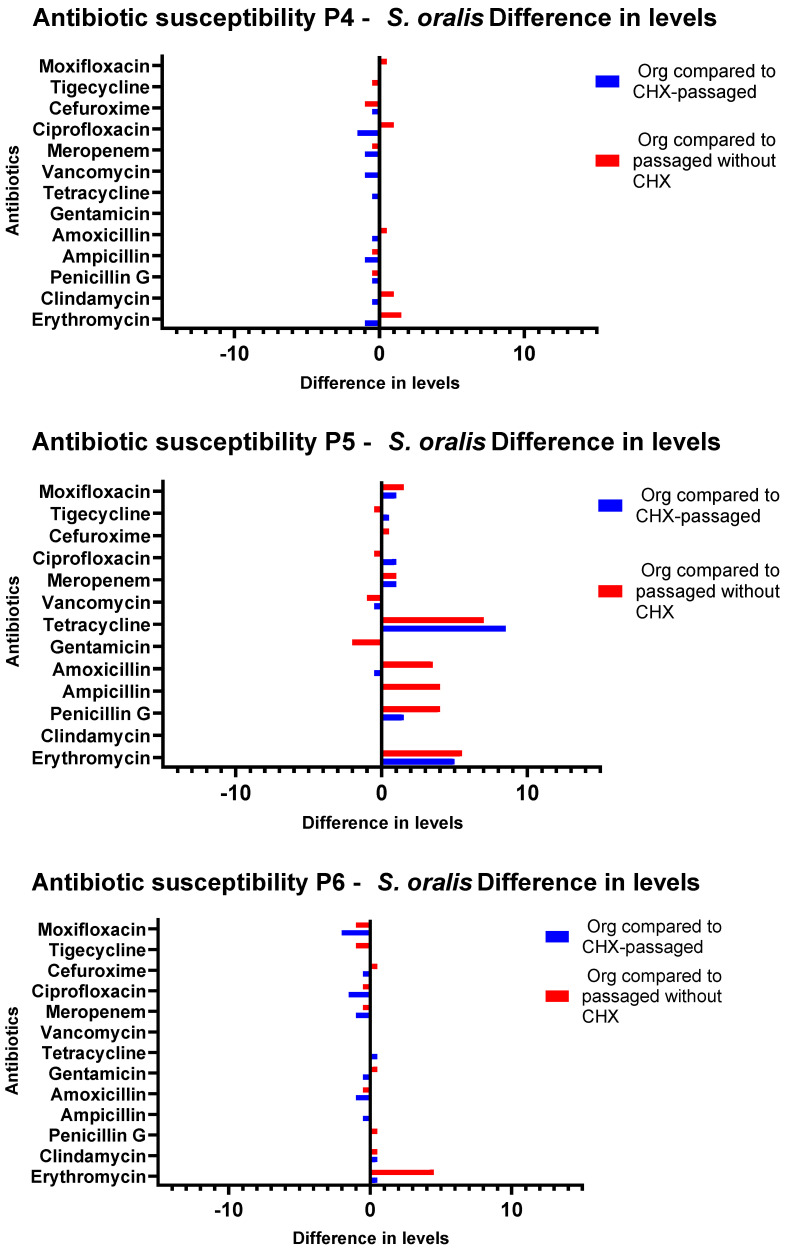
Antibiotic susceptibility testing of *S. oralis* isolates from Subject 4–Subject 6. Graphical depiction of the difference of susceptibilities in levels between the original isolate and the isolates after passage with and without CHX. The levels are drawn on the X-axes, and the antibiotics are shown on the Y-axes. P4–P6: Probands 4–6.

**Figure 7 antibiotics-11-01420-f007:**
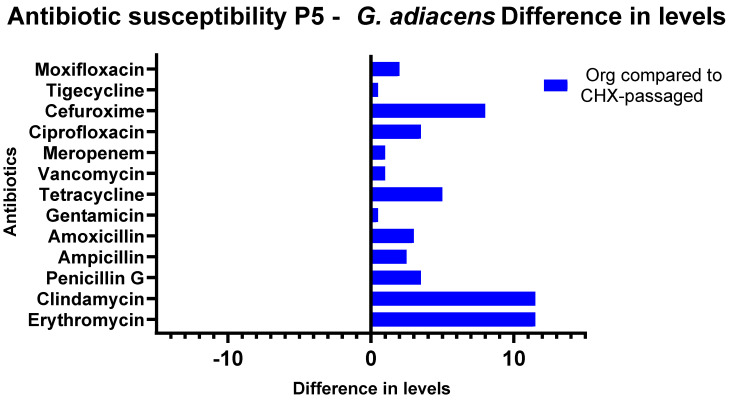
Antibiotic susceptibility testing of *G. adiacens* isolates from Subject 5. Graphical depiction of the difference in susceptibilities in levels between the original isolate and the isolates after passaging with CHX. The levels are drawn on the X-axes, and the antibiotics are shown on the Y-axes. P5: Proband 5.

## Data Availability

The datasets used or analyzed during this study are available from the corresponding author upon reasonable request.

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
