# Peer review of "Antibiotic Resistance of Selected Bacteria after Treatment of the Supragingival Biofilm with Subinhibitory Chlorhexidine Concentrations"

_antibiotics, 2022, doi:10.3390/antibiotics11101420_

Round 1
Reviewer 1 Report
Although the manuscript is well-written and informative, the way that the authors have presented the data is hard to follow as it is not standardized. Furthermore, the results do not support the conclusions in some instances. For example, the authors state that there has been a >4-fold increase in MIC after passage with CHX. However, in most of the S. oralis isolates tested, it has been a 2-fold to 4-fold increase (one tube difference if using standard MIC broth dilutions), e.g. P1 (0.0025 to 0.005%). The only >4-fold increase is in P5 (0.001 to 0.005%?). To this reviewer, the results are not so convincing. The antibiotic susceptibility results are also not standardized with differing X-axes values - what were the actual values (in ug/ml)?
Minor errors:
1. Line 37 - World Health Organization (U.S. English spelling).
2. Line 95 - "... and have not used antibiotics in the last six months..."
3. Line 97 - reduced transport fluid is not hyphenated.
4. Line 131 - "assorted", not "asserted".
Author Response
Reviewer 1
Point 1: Although the manuscript is well-written and informative, the way that the authors have presented the data is hard to follow as it is not standardized.
Response: The results of the data presented were adapted so that all of the figures for MIC, biofilm formation, and antibiotic susceptibility display the same layout and scaling.
Point 2: Furthermore, the results do not support the conclusions in some instances. For example, the authors state that there has been a >4-fold increase in MIC after passage with CHX.
However, in most of the S. oralis isolates tested, it has been a 2-fold to 4-fold increase (one tube difference if using standard MIC broth dilutions), e.g. P1 (0.0025 to 0.005%). The only >4-fold increase is in P5 (0.001 to 0.005%?). To this reviewer, the results are not so convincing.
Response:
The abstract now describes a change of 2 levels up to 4 levels, to represent all of the results. The isolates of S. oralis from P1, P2, and P4 show a 2-fold increase from 0.0025 to 0.005% CHX. The isolates of P3, P5, and P6 show a 4-fold increase from 0.001 to 0.005% CHX. The paper does not report an increase >4-fold, but rather of 2 to 4-fold, and hence the results should now support the conclusions. The authors are aware that the number of levels of adaption seems relatively low (2 to 4-fold) and thus should be discussed carefully. Nevertheless, the authors concluded that the results are reliable because the changes observed are constant for each bacterial strain in a large number of replicates.
Point 3: The antibiotic susceptibility results are also not standardized with differing X-axes values - what were the actual values (in ug/ml)?
Response:
The values of the x-axes were standardized. The representation of the MIC values of the antibiotics in µg/ml is not presentable because the values have a completely different range depending on the antibiotic used. Accordingly, a representation in one diagram is challenging. By displaying the change in steps, all antibiotic tests pertaining to a given bacterial strain can be displayed in one diagram. The actual values of the isolates have now been added as supporting information.
Point 4: Line 37 - World Health Organization (U.S. English spelling).
Response: This has now been corrected.
Point 5: Line 95 - "... and have not used antibiotics in the last six months..."
Response: This has now been corrected.
Point 6: Line 97 - reduced transport fluid is not hyphenated.
Response: This has now been corrected.
Point 7: Line 131 - "assorted", not "asserted"
Response: This has now been corrected.
Reviewer 2 Report
I have read the article entitled "Antibiotic Resistance of Selected Bacteria after Treatment of the Supragingival Biofilm with Subinhibitory Chlorhexidine Concentrations" with great interest and I think it is in principle suited for a publication in the Special Issue “Antimicrobial Strategies against Oral Pathogenic Bacteria and Biofilm”. It is well-written, clearly exposed and well structured. However, I also have some questions and minor comments.
How many strains were identified in supragingival plaque samples?
Is it possible to present results of mass-spectrometric analysis in a Supplementary?
Line 80. Please check reference #24.
Line 80-81. I think it would be more correct to write that "in another study showed negative genotoxic potential for CHX."
Line 121. “This procedure was executed for 10 passages (P1 to P10).” Why are the results in the Figures ## 2, 3 shown only for passages 1-6 for S. oralis and only for P5 for G. adiacens?
Line 177. Please, change S. oralis to Streptococcus oralis.
Figures ## 5, 6. The scale of x-axes should be the same (e.g. from -15 to 15) for all plots.
Author Response
Reviewer 2:
Point 1: How many strains were identified in supragingival plaque samples?
Response: Since the original samples were grown under the selection pressure of CHX, only two species (Streptococcus oralis and Granulicatella adiacens) survived the passage while all other species were eliminated via the ten-day CHX-passage. The large diversity of the original plaque samples was evident when they were grown on the agar plates. However, the microbial diversity of the original plaque samples was not analyzed because it was not the focus of this study.
Point 2: Is it possible to present results of mass-spectrometric analysis in a Supplementary?
Response: The additional results of the mass-spectrometric analysis of the isolates have now been included in the supporting materials.
Point 3: Line 80. Please check reference #24.
Response: References #23 and 24# were in the wrong order due to formatting. This has now been corrected.
Point 4: Line 80-81. I think it would be more correct to write that "in another study showed negative genotoxic potential for CHX."
Response: This has now been corrected.
Point 5: Line 121. “This procedure was executed for 10 passages (P1 to P10).” Why are the results in the Figures ## 2, 3 shown only for passages 1-6 for S. oralis and only for P5 for G. adiacens?
Response: This appears to be a misunderstanding, as the passages were also abbreviated with P1–10 but, as can be seen from the legend of the diagrams, here the P was used to abbreviate proband in the diagram (six probands P1–P6). Granulicatella adiacens was only selected in the sample of proband 5 and hence, G. adiacens was only shown for P5.
Point 6: Line 177. Please, change S. oralis to Streptococcus oralis.
Response: This has now been corrected.
Point 7: Figures ## 5, 6. The scale of x-axes should be the same (e.g. from -15 to 15) for all plots.
Response: The scale of the x-axes was standardized.
Reviewer 3 Report
In this paper, the authors well described the significant untoward effects of the
chlorhexidine digluconate administration (CHX). In fact, they both observed an
adaptation of the microbial strains to CHX and a decreased susceptibility to the
antiseptic itself. The results achieved are important and have a novelty in the point
of the continuous growing awareness of the correct use of antimicrobial compounds
at least to slow down the unavoidable resistance phenomenon. It is a very interesting
paper.
However, there are some issues that should be addressed before publication concerning the statistical analysis. In fact, for figures 1-4 a two-way ANOVA would be advisable to highlight possible significant differences.
Author Response
Dear reviewer, thank you very much for the revision suggestions.
Point 1: However, there are some issues that should be addressed before publication concerning the statistical analysis. In fact, for figures 1-4 a two-way ANOVA would be advisable to highlight possible significant differences.
Response to Point 1:
Well ANOVA with interactions is a good train of thought and we would have liked to calculate this.
However, we have - since the data did not give it otherwise from a distributional point of view - calculated NON-PARAMETRIC.
Since ANOVA is based on the parametric calculation, it is unfortunately not possible in this case.
Round 2
Reviewer 1 Report
Thank you for making the required changes.